# Position: Strong Consumer Protection
# is an Inalienable Defense for AI Safety in the United States

**Serena Booth** [1]

## Abstract

Consumer protection laws are designed to protect consumers from unethical business practices. I argue that these laws serve an emergent dual purpose: consumer protection laws can serve as an inalienable defense for AI safety by incentivizing businesses to design and deploy safer AI systems. This position counters two alternative ideas in AI policy. The first alternative position is that AI safety requires a new set of focused laws to protect humanity's prosperity. Though there are gaps in the existing law and opportunities for improvement, I argue that crafting new, AI-focused laws is both technically challenging and that these laws will be easy to skirt. The second alternative position is that consumer protection is little more than red tape; I argue that existing laws dating back many decades have already reigned in some nefarious business practices related to the development and deployment of AI, and that the litigious society of the United States is well positioned to use consumer protection laws to encourage the development of new guardrails for AI. This paper tours some existing consumer protection laws in the United States and their effects on the development and use of AI systems. This paper also calls to enforce and preserve these laws in a rapidly changing, de-regulatory political landscape.

## 1. Introduction

Seemingly overnight, the threats AI poses to humanity's prosperity—our jobs, our social infrastructure, our education, and even, some argue, our continued existence—went from fringe concerns and science fiction to mainstream scientific discourse (Bengio et al., 2024). In response to this discourse, AI Policy has taken Washington, D.C. by storm: the White House and both branches of Congress have released vague roadmaps for AI legislation (Biden-Harris Administration, 2023; Office of Science and Technology Policy, 2022; The Bipartisan Senate AI Working Group, 2024; United States House of Representatives, 2024). These efforts to regulate AI directly are well intentioned, but in this position paper, I share an optimistic perspective about the state of our existing statute—notwithstanding the current deregulatory emphasis in the U.S. Federal Government. **The existing consumer protection laws of the United States provide many substantial protections for AI safety, and we should preserve, enforce, and strengthen these laws**.

There are two primary alternative points of view to this position. The first is that consumer protection laws provide little protection in the face of highly-advanced AI systems. While I support efforts to regulate AI directly, I broadly find that such efforts are both hard to write down in precise legislative language and comparatively easy to circumnavigate; one basic argument is that there is no consensus on the meaning of "artificial intelligence." Consumer protection, in comparison, is inalienable in its broad applicability and long-standing legislative records, and these existing laws apply to the development and deployment of AI systems. I argue that there have already been many examples of consumer protection laws, many written decades before even the advent of personal computing, that have provided protections to novel applications and developments of AI. The second alternative view broadly takes a dim perspective on consumer protection: this perspective interprets consumer protection as little more than unnecessary bureaucracy and argues that fewer guardrails on markets would result in increased efficiency. This cynical perspective seems harshly distorted from reality: I argue that where consumer protection laws are weak or poorly enforced, the market has not naturally corrected. Moreover, the European Union's AI Act—a momentous law focused on minimizing risk from AI systems—is centered around issues of consumer protection.

This paper takes a tour through some existing consumer protection laws in the United States and discusses the effects these laws have had on AI development and deployment to

[1] Brown University, Providence, RI. This work was done in part while the author was visiting the Simons Institute for the Theory of Computing at the University of California at Berkeley. Correspondence to: Serena Booth <Serena_Booth@brown.edu>.

*Proceedings of the 42nd International Conference on Machine Learning*, Vancouver, Canada. PMLR 267, 2025. Copyright 2025 by the author(s).

date. This paper showcases some of the successful applications of these laws to AI systems, but also some weaknesses of existing consumer protection statute. It calls for strengthening and enforcing consumer protection laws for the sake of encouraging the development and deployment of safer AI systems. This paper focuses on consumer protection in the United States, though many of these insights apply internationally. The selection of laws discussed in this paper corresponds to the jurisdiction of the United States Senate Banking, Housing, and Urban Affairs Committee, where I served as an AI Policy Fellow from 2023 to 2025 to provide technical expertise for the oversight of laws and regulations with equities in AI. While the political climate of the United States has rapidly entered a deregulatory period, studying successful or conceptualized consumer protection laws can help us chart a productive pathway forward in the future.

## 1.1. Scope: Consumer Protection

This paper is scoped to focus primarily on consumer protection—broadly, legislative and regulatory efforts to protect consumers from fraudulent, deceptive, or unethical business practices. Consumer protection is a well-developed area of the law that provides significant protection but still needs refinement in response to new AI technologies, making it rich for discussion. Strong consumer protection is a first line of defense for AI safety: for example, with private right to action, litigation can reign in early appearances of errant capabilities, which in turn appropriately incentivizes AI companies to design safer AI systems. The largest risks AI poses to humanity arguably come from breaking our social norms and compromising our values, hence the substantive line of research into value alignment for AI systems (Amodei et al., 2016). I believe these norm violations and compromised values will appear early in issues of consumer protection, due to the sensitive nature of these applications and consumers' rights. With strong rights and protective institutions, consumers will initiate legal challenges as these violations appear, at least where these violations are perceptible. Naturally, consumer protection excludes many equities of AI safety, as it excludes all military and defense applications, most healthcare applications, and many labor applications—though labor protection and consumer protection are increasingly two sides of the same coin, these areas of the law are treated separately in U.S. governance. These other applications and settings are equally essential for AI safety, and should be carefully considered elsewhere.

## 1.2. Scope: Artificial Intelligence

This paper adopts a broad and colloquial definition of AI, in part from necessity. This definition includes all deep learning methods, all forms of discriminative AI, and all forms of generative AI, including large language models. This broad definition is necessary given the inherent ambiguity: many

corporations claim to be using AI or machine learning, even when they are not or their techniques are simplistic, as a mechanism for driving engagement or investment (Hopkins & Booth, 2021). Interestingly, since it is hard to determine whether companies are actually using AI from the outside, the Securities and Exchange Commission has started cracking down on the fraudulent practice of "AI-washing," where companies overclaim their AI capabilities (Securities and Exchange Commission, 2024).

## 1.3. Scope: AI Safety

AI Safety is an amorphous research area with many different interpretations. This broad area of research covers narrow topics like the off-switch problem—roughly, how can we prevent AI systems from learning how to disable their off-switches (Hadfield-Menell et al., 2017)? Another focus of AI safety is to prevent AI systems from spewing hate speech, as has been repeatedly observed from these systems, with a notable early example being Microsoft's Tay Chatbot that devolved into hateful and offensive speech after just 16 hours on the internet (Miller, 2017). Yet another focus of AI safety research concerns the Chemical, Biological, Radiological, and Nuclear threats these systems pose: e.g., can an AI system increase non-specialists abilities to synthesize biological weapons (Urbina et al., 2022)? Another emphasis is on the gradual disempowerment of humankind, where humans' influence on large-scale social systems is slowly degraded by increased AI reliance (Kulveit et al., 2025).

In contrast, this paper takes a broad view of AI safety—incorporating all aspects of how AI systems will affect human livelihoods, jobs, and economic outlooks. This is also an increasingly common perspective within the broader machine learning research community (Hazra et al., 2025). Consider, for example, the case study of using AI to set prices for goods and services: it is possible that using regret for "willingness-to-pay" as the reward for learning price-setting algorithms could squeeze people into lives of poverty. This is an example of unsafe AI that can be addressed through strong and even technology-agnostic consumer protection laws. I acknowledge that some AI safety concerns will not be directly addressed by consumer protection laws—nonetheless, these laws provide important guardrails that may even prove useful for mitigating even some of the more focused AI safety concerns since these laws necessitate that developers and deployers will more closely analyze their AI systems. Such increased scrutiny may prove beneficial beyond the scope of these laws.

## 2. AI Risks and Consumer Protection

This paper makes a critical assumption about AI safety, which is up for debate and speculation. Namely, I argue that the harms that AI poses to humanity will emerge gradually,

and not overnight. This argument is largely speculative, and there is no consensus within the scientific community. Still, there is some supporting evidence in favor of this perspective. For example, Bubeck et al. (2023) released a controversial paper claiming that GPT-4 presented "sparks of artificial general intelligence," based largely on its capabilities in coding and mathematics—an attempt to observe the emergence of high risk capabilities. This idea—that we can detect sparks of capabilities—suggests that the development of especially threatening AI systems will be gradual. Though the release of ChatGPT has sparked a flurry of economic activity over the last two-and-change years, it and similar AI systems have only modestly affected the labor market (Demirci et al., 2024) and has yet to return on investment (Waters & Bradshaw, 2024); while two years is not long, it is sufficient time for crisis intervention.

Even if the potential harms posed by "superintelligence" or "artificial general intelligence" do appear overnight, there is still substantial cause to carefully review and strengthen consumer protections in the era of AI. Many of the harms these systems are capable of are already present in existing systems—for example, Facebook was found to (illegally) discriminate in delivering advertisements for housing, allowing advertisers to exclude Black and Hispanic users from seeing housing ads (Ali et al., 2019). Consumer protection laws, like the Fair Housing Act that formed the basis for this lawsuit, safeguard against many existing harms, like discrimination. Beyond diminishing existing threats from AI systems, strengthened consumer protection may guard against future threats: for example, stronger data protection standards might be able to prevent future AI systems from collecting the information needed to perpetuate certain types of harms, like those that are enabled by using deepfakes.

## 3. Consumer Protection and AI Case Studies

Consumer protection laws are extremely broad and varied. In the following sections, we will analyze many different consumer protection laws—many dating back decades or even a century—and discuss how these existing laws have affected AI development and usage, and where there are regulatory gaps that should be patched to improve how consumer protection laws can be applied to AI systems.

### 3.1. Algorithmic Collusion and the Sherman Act

RealPage is a company that provides property management software; it is best known for its widely-used algorithm for setting rents. While RealPage has claimed their algorithm uses machine learning and AI, the details of the algorithm remain proprietary and inscrutable (Hughes, 2024). RealPage has advertised to landlords that they could earn between 3 and 7 percent more than the market average if they used their rent-setting software. It is reported that the company largely achieved this by systematically increasing vacancy rates while raising rents across the markets in which they operated; after they acquired sufficient market dominance, RealPage has purportedly been able to raise rents across entire cities beyond the expectations of a competitive marketplace (Vogell et al., 2022). In 2024, the U.S. Department of Justice and attorney generals from eight States filed a legal case against RealPage and many of their affiliate corporate landlords. This case accuses RealPage of artificially inflating rents through collusion in violation of the Sherman Act, an antitrust law from 1890 that prohibits price fixing (U.S. Department of Justice, 2024). While this case remains in limbo in the court system, it reveals several interesting insights into the state of consumer protection, AI regulation, and AI safety.

Though litigation is pending, the actions of RealPage appear almost certainly to have broken the law. As alleged in the case filings, RealPage collected non-public data from landlords, including "a landlord's rental prices from executed leases, lease terms, and future occupancy" (U.S. Department of Justice, 2024). In collating this non-public data from many competitors, RealPage engaged in explicit collusion and price-fixing. While the authors of the Sherman Act presumably imagined collusion taking place in a smoke-filled back room, and though the first general purpose computer would not exist for another 56 years, a reasonable interpretation of the statute nonetheless still encompasses RealPage's rent-setting operation. This is the first lesson for AI safety: technology-agnostic laws that protect consumers can be effectively applied to reign in the power of corporations using modern algorithms and AI systems, even far beyond the imagination of the laws' original designers. Enforcing this law will deter both RealPage and its successors from using similarly illegal algorithmic collusion techniques in the future. Still, the pace of the legal system must improve: the first reporting on RealPage came out in 2022 (Vogell et al., 2022), and—three years later—this technology is still being used by corporate landlords across the nation.

In response to the case of RealPage, some United States Senators proposed a new piece of legislation to explicitly prohibit this practice of algorithmic rent-setting (Senators Welch and Wyden, 2024). While well-meaning, this is a cautionary tale: the judicial system interprets the law through the legislative record, but also through Congress's collective debates and actions. Since this newly-introduced specific legislation addresses an already-illegal action, it could be used as evidence in the judiciary that this cut-and-dry application of the existing Sherman Act statute might not apply since Congress is actively debating whether to support this new legislation. In this manner of introducing legislation that focuses on AI, this introduction of a new law risks undermining the precedent of applying the Sherman Act to this collusive practice of rent-setting with AI.

Nonetheless, the Sherman Act is limited in its applicability for AI systems and needs revising for the modern era. The actions of RealPage are illegal because of the explicit nature of their collusion: they solicited non-public data from ostensibly-competing landlords, and constructed a single, shared algorithm for determining rents and vacancy targets from this non-public data. I argue that RealPage could have achieved these same outcomes and nefarious results *without* using any non-public data. AI is exceptionally good at inferring missing data (Schafer & Graham, 2002)—arguably, pattern recognition of this form is both the main function and main success of AI. A sophisticated predictive model could readily infer whether an apartment is occupied or not based on public data: whether the previous occupants have pulled their creditworthiness in another rental application, whether the apartment is publicly listed for rent, whether the voter registration at the unit has changed, and so on and forth. While such inferences would introduce more error, it is arguably unlikely that this noise would significantly degrade performance of a rent-setting algorithm. In such a setting—if RealPage had instead produced the same service but only used public data—it is much more ambiguous whether it would have violated the Sherman Act. Such tacit algorithmic collusion, in the modern era of AI, is also increasingly plausible (Arunachaleswaran et al., 2024). Supra-competitive prices that outpace reasonable market rates are plausible even when different retailers use different algorithms (Calvano et al., 2020). In response to these threats of supra-competitive pricing, we must strengthen our anti-competitive consumer protection laws beyond the Sherman Act—and there are some emerging proposals to audit algorithms for tacit collusion (Hartline et al., 2024).

The risks to consumers that stem from using AI systems for setting prices also extend beyond the realm of collusion. AI systems, unbridled access to consumer data, and increased market monopolization allow for the increased personalization of prices—which risks exposing consumers to illegal discrimination (Gillis & Spiess, 2019) and sustained, untenable price increases for consumers (Bar-Gill, 2018). The flip side of this coin of setting prices is setting workers' wages. Here, too, there is evidence of emergent so-called "algorithmic wage discrimination," where wages are vary with protected characteristics (Dubal, 2023). Since these issues do not center on collusion, they are protected in part by other laws—like the Civil Rights Act. But these protections are insufficient and have not yet been enforced in these developing settings. The collective risks induced from algorithmic pricing and wage-setting—both the issues of collusion and discrimination—should be studied and addressed holistically by the legislative system.

The discussion of algorithmic price setting and the Sherman Act shows how technology-agnostic consumer protection laws can reign in nefarious AI deployments, but this case study also showcases the gaps in the existing laws, like the lack of coverage for increasingly tacit collusion enabled by the introduction of ever more sophisticated AI systems. This case also showcases how the slow justice process allows harmful AI to proliferate, even when the law has the necessary protections on record in statute.

### 3.2. Explainable AI and Equal Credit Opportunity

The Equal Credit Opportunity Act (ECOA) of 1974 is an example of a consumer protection law that has had a dampening effect on the use of AI in high risk sectors far beyond its original intentions. This law was designed to facilitate the inclusion of women in the financial system: ECOA banned financial institutions from discriminating based on protected characteristics, like sex and marital status, at a time when it was common for unmarried women to be denied credit (Ladd, 1982). Meanwhile, the development of AI was exceedingly nascent in this time—Deep Blue would not be created for over another twenty years (Campbell et al., 2002), and personal computers were far from commonplace.

One of ECOA's provisions, in an effort to mitigate discrimination and increase transparency into the credit approval process, requires that financial institutions must provide an explanation for any adverse action taken in response to a loan application (Maltz & Miller, 1978). An adverse action consists, for example, of denying the request, of offering worse terms than requested, or of similar treatment in which the applicant receives less or worse credit than desired. This explanation requirement is an imperfect mechanism to verify that discrimination did not contribute to the adverse action. This provision, at first blush, is tricky for modern AI systems and deep learning systems to adhere to. The field of Explainable AI has sought to provide explanations for the decisions made by predictive systems; such explanations could in principle allow financial institutions to integrate AI systems into their credit assessment workflows.

Explainable AI techniques, though, have been found to be unfaithful to the underlying decision-making of AI systems (Adebayo et al., 2018; Zhou et al., 2022). Perhaps in response to these concerns, the Consumer Financial Protection Bureau (CFPB) released an interpretation of the existing ECOA law to advise financial institutions that "consumers must receive accurate and specific reasons for credit denials" (CFPB Newsroom, 2023). Effectively, the CFPB stated that deep learning-based systems cannot be used to make credit decisions, especially without significant human oversight. This is another example of how functioning consumer protection laws protect against AI harms: when ambiguous or when challenged by new technologies, these old laws can be appropriately interpreted by regulators. This reinterpretation ability has been diminished by the fall of the Chevron doctrine, as discussed in Section 5

There is a potential downside to this strategy of using consumer protection laws to curb AI development and use. There is some optimism that AI systems can be less discriminatory than human decision-makers (Kleinberg et al., 2018), but through applying the law in this manner we have curtailed attempts to do so. While not entirely a consequence of limiting AI's use in credit assessments, there is another limitation to this application of the law. There is cautious optimism that nontraditional data can increase inclusion in the financial system, if approached carefully (NCLC, 2022)—for example, considering cash-flow can increase financial access for gig workers. Using such nontraditional data is easier with AI systems—so the guardrails of ECOA can be a limiting factor in financial access (FinRegLab, 2023).

Are there solutions to allow industry to innovate and use modern AI while appropriately protecting consumers? One approach is to adopt a "regulatory sandbox," to allow financial institutions to conduct small-scale experiments on consumers. This approach has been adopted in the United Kingdom for FinTech innovation, and has resulted in increased venture funding in these industries (Cornelli et al., 2024). However, it is hard to protect consumers—as the regulatory regime attempts to do—while allowing this type of experimentation in the wild, and such an approach opens up a race-to-the-bottom for consumer protection laws. An alternative blue-sky approach might pursue regulator-industry collaborations to demonstrate the intended increase in financial inclusion and carefully modify the consumer protection laws, like ECOA, where appropriate, but regulator-industry collaborations remain challenging.

This case study of explanations in ECOA demonstrates how the legislative system can guide the design of AI systems. Whether or not a financial institution is using AI, they must be able to provide an accurate explanation—so, this constructs a commonsense requirement for the development of these systems. If AI systems remain inscrutable, they cannot legally be used for this high-risk application. This requirement is not without issue, though: most optimistically, a novel legislative proposal could encourage the use of inscrutable AI that reduces discrimination.

### 3.3. Disparate Impact, ECOA, and the Fair Housing Act

The disparate impact doctrine of the United States, which is present in both ECOA and its housing-focused counterpart, the Fair Housing Act, is a strong ideal, though in practice this doctrine provides insufficient protection for consumers. The idea behind disparate impact is that we can ignore intent when assessing discrimination: policies must not result in unjustified discrimination relating to protected characteristics like race and disability status, even if these protected characteristics are not causal in the decision-making. The strength of this doctrine is in including unintended discrimi-

nation. One weakness of this doctrine is in its permissiveness for "justified" discrimination when discrimination is necessary for preserving the interests of a business.

Disparate impact doctrine has two main implications for the development of AI systems that will be used in any form of credit or housing-related decision-making. The first is an obligation to test these systems for unjustified discrimination. While the need to rigorously test AI systems for disparate impact seems obvious with or without any legal standards, the most famous paper to point out algorithmic bias in deployed systems—Gender shades (Buolamwini & Gebru, 2018)—came out in 2018, half a decade after many of the underlying technologies were invented. Having a legal standard—even a weak legal standard, like disparate impact—compels the developers and deployers of AI systems to test these systems appropriately.

The second implication of disparate impact doctrine is that companies are compelled to search for the *least discriminatory alternative* since, if a less discriminatory alternative can be found without undermining the interests of the business, this means the company broke the law. In practice, the search for least discriminatory alternatives operates a bit like baseline comparisons in academic research. For the sake of compliance with the law, companies document their search for a less discriminatory alternative. But—as is inevitable in the presence of resource constraints and human fallibility—more focus is paid to constructing the discriminatory model of interest, and less attention is paid to the search for a less discriminatory alternative (Pace, 2023). To remedy this, there are some proposed approaches to verifying the existence of a less discriminatory model, at least for a constrained set of alternative models (Gillis et al., 2024).

While disparate impact doctrine is quite weak, it pushes AI developers and deployers to test their systems for emergent discrimination and to search for less discriminatory alternatives. Such testing should be expected as a routine matter in assessing these systems, but it is often neglected due to factors like the rapid pace of development and the difficulty in constructing test data. Disparate impact is also a politically divisive topic; I discuss the current status of this doctrine in Section 5.

### 3.4. Risk Management and Third-Party Authorities for Financial Regulators

One interesting development from the rise of the technology industry is how it has fundamentally altered the fabric of financial services and created many more interdependencies between businesses, small and large. These days, financial institutions routinely rely on third-party vendors—whether for cloud computing services, management software, or AI services (Naimi-Sadigh et al., 2022). The risks that third-party vendors can pose to the financial system are

immense: hackers target third-party vendors, and failures or disruptions cascade across the financial system. Such software failures are common, whether caused by malice or incompetence: in 2024, CrowdStrike's faulty software update prevented consumers from accessing online banking, halted travel worldwide, and may have cost the economy up to $10 billion dollars (Rose et al., 2024).

One of the strongest consumer protections in financial services is that financial institutions are supervised by federal regulators to ensure these institutions are acting appropriately and managing risks to a sufficient standard—for example, by the Federal Reserve or the Consumer Financial Protection Bureau. These supervisory authorities are intensive: regulators can conduct on-site supervision, in which they assess whether the financial institution meets expectations on criteria like the availability of capital or its ability to manage stress tests, as prescribed in the aftermath of the 2008 Subprime Mortgage Crisis (Ryznar et al., 2015). Supervision of financial institutions is, of course, a double-edged sword: while it is in many ways our strongest consumer protection tool, it also massively increases bureaucratic overhead and introduces political liabilities and enhanced risks of corruption in the regulatory regime (Barth et al., 2004).

The introduction of highly-capable AI systems into the banking industry warrants considering whether to expand the role of supervision and stress tests of financial institutions. Since most of these institutions rely on third-party vendors to provide AI-related services, one viable option is to expand supervision down the supply chain. Technically, this authority to supervise third-party vendors already exists for most financial regulators—the exceptions are the Federal Housing Finance Agency and the National Credit Union Administration, and these omissions should be fixed. The question is when and how to invoke these authorities appropriately, given the high costs of doing so. Still, this is an unparalleled opportunity to enforce the safe development and deployment of AI systems, and supervisory authority is one of the strongest mechanisms of oversight available in our current governance structures. In particular, this—to my knowledge—is our only mechanism that could oversee the underlying data on which these systems are trained.

Supervisory authority is a powerful tool for assessing the risks that AI systems pose to consumers and to the financial system, and this authority should be expanded to the financial regulators who currently do not hold it. While invoking this supervisory authority would be costly and should not be done with reckless abandon, this authority nonetheless provides an opportunity to closely scrutinize the AI systems in use at financial institutions. Financial regulators should hire or train in-house AI experts and technologists in anticipation of enacting this authority at some point in the future. One application of this authority could be to conduct

internal audits, as external audits are always limited in their limited visibility to AI systems' designs.

### 3.5. The Rise of AI-Based Fraud and Scams and the Electronic Fund Transfer Act

Fraud and scams are rising dramatically: between 2021 and 2022, fraud and scams increased by 30% in the United States, increasing consumer losses up to nearly $9 Billion dollars per year, as reported by the Federal Trade Commission (Federal Trade Commission, 2023). Fraud and scams are by no means entirely a problem of AI—even old-school check fraud has been increasing dramatically in recent years (FinCEN, 2023). Fraud and scams are increasingly organized: there are many 'scam farms,' where human workers are coerced into conducting scams using the latest available technology ( Agence France Presse, 2025). AI is a potent threat and catalyst for fraud and scams, and requires both significant enforcement of existing consumer protections as well as the addition of new protections.

One of the most alarming threat models enabled by modern AI is as follows: Alice receives a phone call from her relative Bob, and Bob explains that he is in distress—perhaps Bob has experienced a medical emergency, perhaps he's been kidnapped—and he needs immediate access to funds through an electronic transfer. Alice recognizes Bob's voice, and so immediately begins to initiate the transfer of funds to Bob. Of course, Bob is not actually in distress: his voice has been cloned, perhaps from short clips of Bob speaking that are now widely available on the internet (Arik et al., 2018). Worse, still—the scammer might be able to conduct a sophisticated operation, using substantial data about Bob and Alice that might be acquired from a data broker or any other source; they can use this information to be exceedingly convincing in their attack. Alice might hope that the financial system protects her from such crimes—that even if she transfers money to the scammers in an effort to assist Bob, she should be protected by her financial institution.

The Electronic Fund Transfer Act (EFTA) that covers electronic transfers is a tricky piece of legislation. While sometimes quite effective in protecting consumers, to a great extent these protections rely on norms and consumer appeasement from financial institutions rather than the literal interpretation of the statute (Sanchez-Adams, 2024). EFTA decomposes fraudulent transactions into "unauthorized" and "authorized" An unauthorized transaction is one that is initiated without the knowledge of the account owner—think, for example, a hacker who has gained access to your account. An "authorized" transaction, on the other hand, is more akin to a scam: even if it was a fraudulently-induced payment, the owner of the account initiates the transaction and therefore the transaction is authorized. Under EFTA, financial institutions generally have to reimburse consumers

for unauthorized transactions, but not for authorized ones.

The forms of technically-authorized transactions facilitated by AI were unimaginable in 1978 when EFTA written. In the case above, Alice is authorizing a transaction to Bob; she is not authorizing a transaction to the scammer. Is this transaction authorized or not? Under EFTA, it is probably authorized—meaning Alice is on the hook for whatever money she loses in this scam. Perhaps that seems reasonable enough; Alice was duped, but she still initiated this transaction, and perhaps she could have been more cautious in sending funds. I argue that this leaves a large vulnerability for AI systems: namely, financial institutions are necessarily much more sophisticated than consumers in detecting and deterring fraud and scams. In the absence of any skin in the game, financial institutions are not sufficiently incentivized to invest in the requisite detection or prevention mechanisms. We need an update to EFTA—ideally, to hold financial institutions at least in part accountable for all fraud and scams on their platforms—to better protect consumers and to prevent further devastation propagating from AI-fueled scams. The United Kingdom is in the process of introducing a strengthened provision of this form that holds financial institutions accountable barring gross negligence from the consumer (Sullivan, 2024).

Voice-cloning itself is in dubious legal territory, but consumer protection laws also need updating in response to this new technology. Only in 2024—in response to the increasing threat of voice-cloning scams—did the Federal Trade Commission (FTC) release a rule that disallows the impersonation of government and business organizations (Federal Trade Commission, 2024a). Impersonating individuals is not included in the scope of this rule, though the FTC has invited public comment on this addendum and is in the process of rulemaking (Federal Trade Commission, 2024b).

### 3.6. Federal Data Privacy Law

Section 3.5, and the increasing prevalence of deepfakes and highly-targeted, AI-fueled scams, raises an obvious question: why is there no comprehensive data privacy law in the United States? The vulnerability for advanced AI systems is clear: with unrestricted access to data on consumers, these systems are able to manipulate, to collude, and to deceive ever more easily—all significant concerns from an AI safety perspective. Still, there is a $200 Billion dollar industry for data brokers that collect and sell information on individuals and groups (Lazarus, 2019); it is even possible and legal to purchase data on United States servicemembers (Sherman et al., 2023). The need for a new data privacy law is hardly controversial: over the years, support has spanned the bipartisan, bicameral ideological gamut from House Member Patrick McHenry (R-NC) (House Financial Services Committee, 2023) to Senator Marsha Blackburn

(R-TN) (Sen. Blackburn, 2024) to Senator Maria Cantwell (D-WA) (House Energy and Commerce Committee, 2023). The politics of data privacy warrant discussion, especially given the high stakes as AI becomes increasingly agentic.

First, there are some existing data protections in the United States. These data protections range from targeted types of data—for example, Health Insurance Portability and Accountability Act (HIPPA) (Act, 1996) provides protections for medical data while the Gramm-Leach-Bliley Act provides at least *some* protections for financial data (Gramm, 1999), though this latter act is focused mostly on disclosures of sharing data and less on preventing the inappropriate sharing of such data. Given the dearth of federal leadership on data privacy, individual States have started to draft or enact privacy laws. Most famously, California created the CA Privacy Protection Agency (CCPA), and 19 other states have followed suit—including Colorado, Connecticut, Texas, and Utah, again spanning a wide ideological gamut (Ajayi, 2023). States taking action has created a new data privacy regime through a spillover effect: in practice, many corporations adhere to the CCPA standard because it is burdensome and expensive to create different data policies for consumers from different regions (Tran et al., 2024).

While States taking action to protect their consumers' privacy is commendable, it has led to further complications in the United States Congress' efforts to draft and pass a comprehensive data privacy law. There are two seemingly irreconcilable points of debate on data privacy. The first is that of preemption (Tran, 2022): now that States have passed privacy laws, a federal law can serve either as a privacy 'floor' or 'ceiling', depending on whether it usurps the State laws. In general, Democrats favor the former—they want the federal standard to be the minimum bar, but to allow States to write further restrictive legislation. In general, Republicans want the latter—they want the federal standard to be the ceiling, because it is burdensome for business to comply with individual State laws. The second point of contention is the idea of private right to action (Scholz, 2021): Democrats, broadly, want to allow individuals who are harmed by businesses' privacy practices to be able to sue directly; Republicans, on the other hand, are concerned about wasteful litigation and protest private right to action.

The link between data privacy and AI development and deployment is intimate. Many of the concerns in AI safety center mass surveillance and manipulation of the peoples (Bengio et al., 2024). Leaving data unprotected—for any human or AI system to freely access or purchase—is a vulnerability in our societal fabric. This vulnerability has been repeatedly manipulated by malicious actors, and the risks have been documented over and over again. Using legally-acquired data, researchers were able to expose secret financial regulator supervisory activities (Gerken et al., 2024), and a

teenager shared the details of servicemembers' locations on foreign military bases on social media based on their fitness tracking activity (McKenna et al., 2018). In the quest for data privacy regulation, I argue that private right to action on data privacy is an essential step for AI safety: litigation will push businesses to resolve these vulnerabilities and to in turn make progress in the safer design of AI systems.

## 4. Alternative Views

We have now discussed many consumer protection laws and their impacts on AI development and deployment. I assert that the need to focus on and develop consumer protection laws further is apparent through this tour of the legislation—but there are opposing views. The first point of opposition concerns the general focus of consumer protection laws. Instead, some believe we should focus very explicitly on designing legislation to regulate AI. This perspective is not in total opposition to strong consumer protection work, and it carries unique challenges in implementation. The second point of opposition is more an affront to the ideals this position paper espouses: some argue that consumer protection is simply not worth the bureaucratic overhead, and that these laws should be abandoned.

### 4.1. Forget Consumer Protection; Regulate AI Directly

A common point of view is that AI needs designated laws to control and govern its development and use. A stronger version of this point of view might argue that there is little point in focusing on general consumer protection laws, and that all regulatory and legislative efforts should be directed specifically towards controlling and governing AI, especially highly-capable AI systems. Although I find there to be value in some attempts to regulate AI directly, there are many pitfalls in this approach. The easiest argument against AI-specific laws is that the very term "artificial intelligence," and especially its eccentric cousin, "artificial general intelligence," lack a reasonable definition and even a semblance of consensus from the academic community (Perkel, 2024).

In an effort to construct precise legal language, the Biden-Harris Administration Executive Order 14110 first directed the Secretary of Commerce to precisely define an "AI model [that has] potential capabilities that could be used in malicious cyber-enabled activity," and offered an interim definition of any system trained with a compute budget of at least $10^{26}$ flops (Biden-Harris Administration, 2023). While this threshold was selected to be significantly more than the existing foundation models of the time, it is still exceedingly arbitrary: it is unclear if such a large model would actually pose the type of imagined malicious cyber-activity threat, and it is also unclear that a smaller model would not pose such a threat. Deferring responsibility to the Secretary of Commerce is also troubling. Under a friendly

administration—for your personal definition of friendly—this approach is flexible and preferential. But, under an adversarial administration, this approach is dangerous: the Secretary of Commerce has been granted the power to define "AI model" as they see fit, which could effectively be used to control all technology. It is easier to precisely define consumer harms, and to task federal agencies with protecting consumers rights, than it is to define dangerous AI models.

There are valiant ongoing efforts to address these issues with explicit AI regulations, and there are many compelling initiatives, like, for example, the works of the new AI Safety Institutes that are appearing around the world and are mostly focused on benchmarks and assessing the capabilities of AI systems (Araujo et al., 2024). Although there might be some overlap, such efforts are in principle complementary to strong consumer protection laws—but, for consumers and in service of AI safety more generally, existing laws like consumer protection laws must apply to AI systems, and there should be no AI exemptions.

### 4.2. The Case Against Consumer Protection

Support for consumer protection laws are by no means universal. These laws are undeniably burdensome for corporations: they often must undertake extensive and expensive efforts to comply with consumer protection laws, and their risks and liabilities increase with stronger protections. Moreover, these strong laws introduce risk surface area for the problem of regulatory capture, wherein corporations gain undue influence over regulatory agencies and work to craft the law and enforcement of their law to serve their business interests—for example, working to stifle competition.

Opponents of strong consumer protection laws argue that consumers will naturally choose to spend money or otherwise associate with businesses that protect their interests, and so consumer protection laws add superfluous, unnecessary, and potentially risky bureaucracy. So, after some time for adjustment, ultimately businesses will adopt the standards needed to protect consumers without the burdensome regulatory compliance or costs of lawsuits. I oppose this position on three grounds, and I believe it has significant implications for AI safety.

First, assuming that businesses indeed generally stabilize to protect consumers, this position allows some consumers to be significantly hurt without recourse through this process—a risk I find unacceptable. Second, it is unclear whether the free market would sufficiently organize to protect consumers: for the case of infringements on reasonable consumer privacy rights, these somewhat-invisible harms have gone unaddressed for decades, both in the free market and in federal law. Third, consumer protection laws are critical for establishing guardrails in service of AI safety. As argued throughout these piece, existing consumer protection

laws dating back many decades have already started to contribute to the design of safer AI systems. Going forward, we must lean into designing smart consumer protection laws that minimize the compliance burden while encouraging the development of safer AI systems.

## 5. Parting thoughts on Consumer Protection and AI Safety

Consumer protection laws are by no means the be-all and end-all of AI policy or AI safety. They are, nonetheless, an inalienable defense—these laws are broad, far reaching, and difficult for businesses to avoid encountering as they develop and deploy AI systems. In the United States, these consumer protection laws have been developed over the last many decades, but there have been few changes in recent years to specifically address the development of AI systems. In the European Union, the landmark EU AI Act is an entirely new consumer protection law that focuses on AI specifically, and uses a risk-based approach to assess how intense the regulatory response should be. Assessing the benefits and drawbacks of each of these approaches can guide peer nations in their approach to regulating AI; this assessment can also assist the United States in rebuilding and reforming its consumer protection laws in the future.

But this inalienable defense requires maintaining, developing, and maturing as AI systems continue to progress. In the United States, one obstacle to this continued enforcement came in 2024 with the fall of the Chevron doctrine (Leason & Martin, 2024). From the Supreme Court's decision in 1984 until 2024, this doctrine prioritized executive agencies' interpretations of ambiguous laws. This was a useful tool in taking laws that were written before the advent of new technology—like personal computing or AI—and reinterpreting the intent of these laws in these modern contexts. The new government administration in the United States has also been working to undermine existing consumer protection statute. For example, many of the laws discussed in this work are enforced through the Consumer Financial Protection Bureau (CFPB), which the administration attempted to curtail one month into their leadership (Wamsley, 2025). The administration has equally tried to curtail the use of disparate impact doctrine (E.O., 2025); disparate impact doctrine is one of our strongest tools for curtailing discrimination in spite of the use of inscrutable AI systems.

The next frontier of AI development is the quest to build agentic systems; such agentic systems bring many new risks, and supercharge existing risks like those of AI-enabled fraud and scams (Yohsua et al., 2024). As agentic systems come into existence, consumer protection laws remain. There will be new legal challenges and new interpretations of existing statute, but still many of these same provisions can be used to hold the corporations designing and deploying agentic AI systems to account. Continued enforcement and proactive oversight of AI developments are essential to ensuring these consumer protection laws continue to serve the people the age of AI. In the quest for artificial intelligence, artificial general intelligence, or superintelligence, and in the fight for AI safety, we must not forget: existing laws apply.

## Acknowledgements

While this work does not reflect the views of my former employer, I am extremely grateful to my former colleagues in the U.S. Senate for teaching me about the importance of consumer protection and economic policy for AI governance. I am also grateful to the reviewers, especially the one anonymous reviewer who deeply engaged with this work and pushed me to improve it significantly. This work was supported in part by the Survival and Flourishing Fund.

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
