# OpenReview forum: "Position: Strong Consumer Protection is an Inalienable Defense for AI Safety in the United States"
_ICML.cc/2025/Position_Paper_Track — ICML 2025 Position Paper Track poster_

### Official Review · Reviewer_DDka · 2025-03-14

**Significance:** 3
**Argument Clarity:** 2
**Rating:** 3
**Confidence:** 4

**Questions:**

No major questions

**Discussion Potential:**

3

**Paper Summary:**

Core contribution: The paper discusses the applicability of consumer protection law as a meaningful accountability mechanism for AI (i.e. more than just red tape), which does not require the introduction of bespoke AI regulation.

Strengths:
- I found the early sections to provide good grounding for the average ML reviewer to be acclimated with consumer protection; I think more could be said (e.g. specific cases, enforcement actions, even the relevant agencies) and I would allocate space to this, while being cognizant of page limits.
- I appreciated the case studies.
- I appreciate the interdisciplinary contribution and will push against other reviews that implicitly try to argue this isn't a "ML" contribution, as I see it as the clear benefit to the field for subjects like this to be discussed in ML venues and not merely delegated to law/policy/up-and-coming interdisciplinary venues.

Edit: I improved my score from a 2 to a 3 following author response.

Weaknesses:
- The logic for selecting the case studies seems pretty unclear to me.
- While it makes sense the case studies are heavily abbreviated from the level of rigor one would see in a law review article, I don't think the cases really articulate the key learnings well enough. Largely, they come off as a somewhat relevant descriptive exercise, without any clear proof or learning to give confidence to the overarching position
- Overall, I felt the work did not really argue the core position in the body, despite the framing of the intro. To be honest, I am not fully convinced the majority (Sections 2-5) actually really constitute a true position paper.
- Given the entire frame of drawing upon consumer protection, I found the paper to be generally less well-cited than I expected. I don't have specific references that come to mind, but the body of legal scholarship on consumer protection is vast and doesn't feel like it is being seriously integrated into the piece, let alone being made more accessible to the ML community.
- The logic of the EU AI Act is grounded in product safety: I would have expected more serious engagement with this, even if the piece is largely set in a US legal frame.


More general concern: In the current political climate, where federal agencies are being hamstrung severely, how is this a tenable position if there are far few government employees and severely reduced enforcement capacity? I worry that this policy position, while advancing sensible logics and broadly-agreeable ideas of applying existing authority, while fine as a divorced intellectual ideal, has little feasibility at present. Of course, I don't think this prohibits accepting this paper, and certainly it is not the author's job to control the political environment, but I do think this needs to be seriously factored in given the paper aims to discuss a very active/timely subject of AI policy.

**Position:**

Yes

**Position In Title:**

Yes

**Related Work:**

1

**Strengths And Weaknesses:**

Covered in previous section

**Support:**

2

---

> ### Author Rebuttal · Authors · 2025-03-31
>
> Thank you for your thoughtful review. We appreciate that you value this work for the ML community, and we will incorporate as much of your feedback as we can.
>
>
> **Selecting the case studies:**
>
> These case studies concern consumer protection issues in the jurisdiction of the US Senate Banking, Housing, and Urban Affairs Committee (https://www.banking.senate.gov/about/jurisdiction). We will add this detail to the paper; it was omitted to avoid potential anonymity issues.
>
>
> **Level of rigor compared to a law review article, consumer protection citations:**
>
> A goal of this position paper is to engage ML and AI researchers in the broad design of AI policy. As such, many of the citations in this paper are to regulatory directives from federal agencies instead of legal academic discourse.
> In this way, the paper is distinct from a law review article, which tend to focus narrowly on a piece of law and its implementation. Instead, this paper seeks to discuss the regulatory structure around a significantly larger set of laws that guide the development and deployment of AI through their interplay.
> Still, we agree that the reader could benefit from additional positioning within the broader legal literature.
> We will add the following citations and welcome additional suggestions:
> * Bar-Gill, Oren. "Algorithmic price discrimination: When demand is a function of both preferences and (mis) perceptions." Chicago Law Review.
> * Gillis, Talia B., and Jann L. Spiess. "Big data and discrimination." U. Chicago Law Review
> * Bruckner, Matthew Adam. "The promise and perils of algorithmic lenders' use of big data." Chi.-Kent L. Rev.
> * Judge, Kathryn. "Fragmentation nodes: a study in financial innovation, complexity, and systemic risk." Stan. L. Rev.
> * Dubal, Veena. "On algorithmic wage discrimination." Columbia Law Review.
> * Mahoney, Peter E. "The End(s) of Disparate Impact: Doctrinal Reconstruction, Fair Housing and Lending Law, and the Anti-Discrimination Principle." Emory LJ.
>
>
> **Arguing the core position:**
>
> A mainstream position in AI policy and especially AI safety is that we need new laws for AI (Sec 4.1). In contrast, this position paper argues that existing US regulatory structure is largely sufficient and provides a strong backbone for considering new regulations for AI. The case studies serve as evidence for this position. Still, we see your point, and so we will add a statement at the end of each case study that clearly refers back to the main position to help the reader. To give a concrete example, for Section 3.1 (‘RealPage’), an effort to write new laws is Congress’s “Preventing the Algorithmic Facilitation of Rental Housing Cartels Act,” that seeks to address services that allow landlords to collude to set prices via algorithms (https://www.welch.senate.gov/welch-and-wyden-introduce-legislation-to-crack-down-on-companies-that-inflate-rents-with-price-fixing-algorithms/). This proposed legislation focuses on the role of AI/algorithms in collusive rent-setting. In doing so, it both undermines the precedent of applying the Sherman Act to this collusive practice and it creates new fragile legislation.
>
> **EU AI Act:**
>
> We scoped this paper to only focus on US consumer protection policy, and due to our specific expertise we are apprehensive about overstepping in discussion of EU policy. We will add a point to Section 5 of this paper to explicitly invite future work to compare the EU and US policy frameworks, hoping that this addition will invite discussion and debate at ICML.
>
> **Political climate, destruction of US regulatory agencies and enforcement of existing laws:**
>
> This is a fair concern. Our position is that, despite ongoing changes, US regulatory structures for the past many years (over a century for the Sherman Act, half a century for ECOA, and so on) have been strong and included many technology-agnostic laws that apply to AI systems and remain worth discussing. US politics is volatile and quickly changing, so arguments showing the strength of a particular regulatory structure may influence future changes. To some, such a paper that reviews past successes might be considered timely to help provide arguments for the defense of previously existing structures. Further, other nations might be influenced by the past successes of US regulatory approaches.
>
> Our current moment presents the AI community with a time to reflect upon what parts of these consumer protection agencies and which laws have been most useful for safe AI development and deployment, perhaps to support advocacy for efforts to maintain or rebuild future regulatory structures. This paper invites this discussion. We will add a section to explicitly discuss the current political climate and the aims of this paper.
>
> A more minor note: this paper was submitted on Jan 31st, before which no actions against such regulatory structures had yet taken place. The proposed added discussion will provide needed context for this developing political situation.

---

> > ### Comment · Reviewer_DDka · 2025-04-03
> >
> > Thanks for engaging and writing an excellent response. One often has little reason to have faith in the review system advancing science when reviewing for ML conferences, but your response is cogent and useful. Congrats
> >
> > - Not sure why what you said would raise a potential confidentiality issue, but at any rate, providing a rationale for why these are well-chosen would be good. With that said, the rationale you provide doesn't seem that compelling as much as descriptive/factual?
> >
> > - Law review comparison: fair, I think your updates seem to be improvements, and I guess my comment on this reduces to "it felt like you could do better", so thanks for considering
> >
> > - Core position: Sure, I can buy that, I found the sharpness you wrote on RealPage in the rebuttal to be more evocative and would love to see something like this directly appearing in the paper
> >
> > - EU: Fair, assumed as much, but I think the reality is you are also writing to the ML community that has experience is not 1, but 0, bodies of law and governance and ... on average. I think its important to talk about the EU AI Act some as the world's first flagship set of laws on the books, especially as regulation in the US may be hamstrung by everything happening under the current federal admin.
> >
> > - What to make of current US status quo: Sure, what you say is very reasonable and sensible, and I mean its not like we stop doing good scholarship given the crumbling federal institutions (and I agree this was submitted prior to some of the key recent developments). At any rate, anything you can usefully write that engages with this reality is very pragmatic and practically useful.

---

### Official Review · Reviewer_zNdy · 2025-03-15

**Significance:** 3
**Argument Clarity:** 3
**Rating:** 3
**Confidence:** 4

**Questions:**

N/A. See weaknesses.

**Discussion Potential:**

3

**Paper Summary:**

This position paper argues that robust consumer protection laws constitute a foundational and inalienable mechanism for ensuring AI safety. Rather than relying exclusively on nascent, AI-specific regulations—often difficult to define precisely and vulnerable to circumvention—the author posits that existing consumer protection frameworks offer immediate, enforceable, and broadly applicable safeguards against the potential harms of advanced AI systems.

**Position:**

Yes

**Position In Title:**

Yes

**Related Work:**

2

**Strengths And Weaknesses:**

Strengths:
A tour-de-force writing: elucidates clear positions arguing how existing consumer laws can be directly applied to AI. A multidisciplinary view that will greatly benefit the ML community, the major readers of ICML papers.

Weaknesses:
While the paper greatly talks about applying existing consumer laws to AI products, the paper could have benefitted more if there were certain recommendations, especially for the AI makers. Most of the current recommendations are for governments or policy bodies; however, I feel direct recommendations to AI might have enhanced the shared prosperity of AI builders and AI consumers. This leads to several questions:
1. Can AI makers do something related to training data usage? How does consumer law help to make decisions in the pre-training pipeline?
2. Similarly, for post-training, how do available laws help AI makers to make better decisions?

**Support:**

4

---

> ### Author Rebuttal · Authors · 2025-03-31
>
> Thank you for your review. We appreciate that you noted this paper presents a multidisciplinary perspective of value to the ICML community.
>
>
> The current regulatory structure is quite cat-and-mouse in its approach to managing innovation and harms, so your comment on how to incite shared prosperity between AI builders and AI consumers is apt.
>
>
> One interesting aspect of consumer protection law as related to pre-training concerns data privacy. As a community, we should ask: are there data these systems simply should not have access to as a matter of privacy? Copyright law is also interesting, but outside of the scope of consumer protection (and a lot of ink has been spent on this already: e.g., Henderson, Peter, et al. "Foundation models and fair use." Journal of Machine Learning Research 24.400 (2023): 1-79.). Available laws provide guardrails and clear rules for AI makers. For example, banks and financial institutions know that they cannot legally develop AI systems to make credit decisions without significant human oversight and accurate explanations. Guardrails are often cited as an effort to thwart innovation, but it can also go the other way: having rules about how AI can or cannot be developed or deployed gives companies confidence in their ability to innovate. It is worse for companies if the law changes in response to their efforts, potentially invalidating collected data, data use, and trained models.

---

### Official Review · Reviewer_D4UC · 2025-03-19

**Significance:** 2
**Argument Clarity:** 3
**Rating:** 2
**Confidence:** 3

**Questions:**

See the weaknesses.

**Discussion Potential:**

2

**Paper Summary:**

This paper argues that consumer protection laws are a key defense for AI safety, providing an existing legal framework to regulate AI harms. Through case studies on algorithmic collusion, AI-driven discrimination, and fraud, it demonstrates how these laws already impact AI governance.

**Position:**

Yes

**Position In Title:**

Yes

**Related Work:**

3

**Strengths And Weaknesses:**

Strengths
- This paper discusses AI safety from the perspective of consumer protection laws, offering a novel angle. The writing is clear and well-organized, and the author effectively uses case studies to provide empirical support.

Weaknesses

My main concerns about this paper are as follows:

- Overreliance on U.S. law – The author should briefly discuss legal frameworks in other regions. The current analysis is heavily U.S.-centric, which may introduce bias.
- Consumer protection laws are reactive rather than proactive – They typically address harms after they occur. The paper could explore how to make these laws more forward-looking in the context of rapidly evolving AI risks (e.g., AI autonomy, systemic bias).
- Limited impact on the AI safety community – Based on the current discussion, I believe the paper’s influence on AI safety discourse is somewhat limited. Can the author further elaborate on the significance of this work to the AI community?

**Support:**

3

---

> ### Author Rebuttal · Authors · 2025-03-31
>
> Thank you for your review. We appreciate that you noted this paper offers a novel angle on the AI policy debates.
>
>
> Regarding your concerns on overreliance on US law:
> This paper is scoped to only focus on US law, as this is our expertise. This is stated in the last sentence of the introduction, “This paper focuses on consumer protection in the United States, though many of these insights apply internationally.” To mitigate this concern, we propose changing the title of the paper to further emphasize its US focus. For example, the title could be: “Position: Strong Consumer Protection is an Inalienable Defense for AI Safety in the United States Legal Code”. As we are apprehensive about discussing legal frameworks in other regions because we are not experts there, we prefer to add an invitation for community discussion with other regions’ legal approaches, as noted to reviewer DDka, to spur future work.
>
>
> Reactive rather than proactive:
> This is an excellent point, and we will add a discussion of this limitation in using the existing legal structure for consumer protection to guard against rogue AI development. The law is not always reactive, though. For example, ECOA’s provision about requiring accurate explanations-and thereby blocking the use of AI in high-stakes credit decisions-is proactive. Enforcement is retroactive, but the banks and financial institutions are sufficiently weary of breaking the law that in most cases they simply will not do this.
>
>
> AI safety:
> This position paper implicitly assumes that questions of AI safety for the wider ICML community are broad (as reviewer zNdy notes), and cover how AI systems will affect human livelihoods, jobs, and economic outlooks; consumer protection law is one part of this. For example, on using AI to set prices: it is possible that using regret for “willingness-to-pay” as the reward for learning price-setting algorithms squeezes people into lives of poverty. This is an example of unsafe AI that can be addressed through strong and even technology-agnostic consumer protection laws. We admit that other AI safety concerns are more specific and a narrower AI safety community exists; we will add a note clarifying this distinction, e.g., consumer protection laws do not address the off switch problem.

---

### Official Review · Reviewer_HSE3 · 2025-03-22

**Significance:** 4
**Argument Clarity:** 3
**Rating:** 4
**Confidence:** 5

**Questions:**

are there any existing or ongoning work on applying those laws into AI safety?

**Discussion Potential:**

3

**Paper Summary:**

This paper tries to tackle the problem of AI safety risk to consumer through the opinion that "Strong Consumer Protection is an Inalienable Defense for AI Safety". This paper provides an overview of existing US laws to protect consumers that existed for hundreds of years before the creation of computers and can be used to protect consumers now with the emergence of AI. In general, this is a very valuable call for using existing laws to regulate AI, especially considering the difficulty of passing new laws for any purposes.
## update after rebuttal

**Position:**

Yes

**Position In Title:**

Yes

**Related Work:**

3

**Strengths And Weaknesses:**

**strengths**:
This paper uses a lot of evidence from existing Consumer Protection Laws to argue that they can also be used for AI regulation.
For example, the "on-site supervision" and "only mechanism that could oversee the underlying data on which these systems are trained"
"some existing data protections in the United States."
"financial institutions are necessarily much more sophisticated than consumers in detecting and deterring fraud and scams."
...

I am not an expert in US laws, but I found this to be a promising direction to release the burden of AI safety posed on existing techniques.
I agree with many of the mentioned examples, though I do not know whether it's legal or not to apply those laws in AI.

**Weaknesses**:
It would be great if the authors could also raise some potential concerns or obstacles of enforcing those laws into AI safety to protect consumers.

**Support:**

3

---

> ### Author Rebuttal · Authors · 2025-03-31
>
> Thank you for your review. We appreciate that you noted the value in applying existing laws to regulate AI and that you think this a promising direction to explore.
>
> Regarding your statement “I do not know whether it’s legal or not to apply those [existing laws] to AI”:
> Typically, applying existing laws to AI systems is straightforward, and almost all of the law is technology agnostic. For example, the CFPB has been extremely clear that existing laws apply to AI (e.g., https://www.meritalk.com/articles/cfpb-director-existing-laws-must-be-used-to-mitigate-ai-dangers/).
> Occasionally, this is more complicated: when these protections are regulations (or, interpretations of the law created by regulatory agencies), these regulations can be more easily challenged in court. For example, regulators have just lost their prioritized position in legal standings: the US Supreme Court struck down Chevron deference in 2024. We will add discussion of this distinction.
>
> Regarding concerns and obstacles to enforcement:
> This also relates to a point raised by reviewer DDka about the general political climate in the United States. We will add a section discussing the resilience (and/or fragility) of these regulatory agencies. This will include their susceptibility to manipulation under political changes and their varying ability to regulate AI in a technology-agnostic way under changes in legal standing.

---

### Decision · Program_Chairs · 2025-04-30

**Decision:**

Accept (poster)

**Comment:**

The paper has three positive reviews, including a champion ("accept"), and I would like to recommend the paper for acceptance.